# Template-controlled on-surface synthesis of a lanthanide supernaphthalocyanine and its open-chain polycyanine counterpart

Qitang Fan[1], Jan-Niclas Luy[1], Martin Liebold[1], Katharina Greulich[1], Malte Zugermeier[1], Jörg Sundermeyer[1], Ralf Tonner[1]* & J. Michael Gottfried [1]*

Phthalocyanines possess unique optical and electronic properties and thus are widely used in (opto)electronic devices, coatings, photodynamic therapy, etc. Extension of their $\pi$-electron systems could produce molecular materials with red-shifted absorption for a broader range of applications. However, access to expanded phthalocyanine analogues with more than four isoindoline units is challenging due to the limited synthetic possibilities. Here, we report the controlled on-surface synthesis of a gadolinium-supernaphthalocyanine macrocycle and its open-chain counterpart poly(benzodiiminoisoindoline) on a silver surface from a naphthalene dicarbonitrile precursor. Their formation is controlled by the on-surface high-dilution principle and steered by different metal templates, i.e., gadolinium atoms and the bare silver surface, which also act as oligomerization catalysts. By using scanning tunneling microscopy, photoemission spectroscopy, and density functional theory calculations, the chemical structures along with the mechanical and electronic properties of these phthalocyanine analogues with extended $\pi$-conjugation are investigated in detail.

---

[1] Department of Chemistry, Philipps-Universität Marburg, Hans-Meerwein-Straße 4, 35032 Marburg, Germany. *email: tonner@chemie.uni-marburg.de; michael.gottfried@chemie.uni-marburg.de

**P**orphyrins, the "colors of life," play important roles in many biological processes[1–3]. The planar porphyrin macrocycle consists of four pyrrole subunits linked by methine groups and bears an 18 π-electron-conjugated inner ring system, which shows a strong absorption band in the visible region, the Q-band, corresponding to the π-π* transition[4–6]. This band determines the optical and electronic properties that are relevant for applications in coating, optics, and semiconductor technology[7]. Gaining control over the various physical properties of porphyrinoids thus requires the size-tuning of the π-conjugated chromophore. Extension of the π-electron system results in a bathochromic shift of the Q-band. This extension can be achieved by increasing the number of annulated peripheral benzene rings or the number of pyrrole subunits. The latter approach has inspired the chemistry of expanded porphyrins[6,8–10], in which extended porphyrinoid macrocycles with intriguing physical properties, such as pentaphyrin, hexaphyrin, and heptaphyrin, have been synthesized[11–13]. This approach paves the way towards sophisticated molecular materials, in particular infrared-absorbing chromophores and third-order nonlinear optical materials[14,15], which are interesting for various applications in optoelectronics, optical limiting[16], digital data storage[17], and photodynamic therapy[18–20].

Phthalocyanines (Pcs) are benzo-annulated tetra-azaporphyrins, in which the meso-methine units (CH) in porphyrin are replaced by nitrogen. Due to this nitrogen doping, they possess modified properties compared to their porphyrin counterparts. Therefore, changing the number of π-conjugated isoindoline units in Pcs can produce molecular materials with properties that cannot be achieved by the corresponding expanded (or contracted) porphyrins. While the synthesis of contracted boron sub-Pcs with three isoindoline units is well established[21], the development of expanded Pcs is impeded by the limited synthetic possibilities[22–24]. The only known example is uranyl-superphthalocyanine SPcU(O)₂[25,26], comprising five isoindoline units, which has been synthesized from phthalonitrile in the presence of large uranium template ions in solution. Apart from this unique case, the reaction of phthalonitriles with other large metal ions such as lanthanide ions typically results in the formation of bis(phthalocyaninato) double-decker complexes rather than five-fold symmetric single-decker super-Pcs[27–29]. Similarly challenging is the synthesis of isoindoline polymers, that is, the open-chain counterparts of Pcs. Typically, Pc macrocycles are the thermodynamically favored products of metal-catalyzed cyclizations of phthalonitriles in solution[30]. Poly(isoindolines) appear as poorly characterized by-products in the synthesis of poly (Pcs) or poly(triazines)[31–34]. Isolation and characterization of well-defined poly(isoindolines) has not been achieved; only isoindoline dimers or trimers have been reported as intermediates in the synthesis of Pcs[35,36]. Related polymers from aliphatic dinitriles have only rarely been reported[37].

In recent years, it has been demonstrated that the co-deposition of *ortho*-dicarbonitriles with metal atoms (manganese[38,39], iron[40]) on surfaces can result in macrocyclization and formation of regular Pcs. Expanding on this approach, we employ differently sized metal templates (iron, gadolinium, and a silver surface) for the synthesis of iron-naphthalocyanine (Fe-NPc), the expanded gadolinium-supernaphthalocyanine (Gd-SNPc), and their unique open-chain counterpart poly(benzodiiminoisoindoline) (here named "polycyanine") from a 5,5,8,8-tetramethyl-5,6,7,8-tetrahydroanthracene-2,3-dicarbonitrile (ADN) building block on a Ag(111) surface (Fig. 1a). The general concept behind this approach is illustrated in

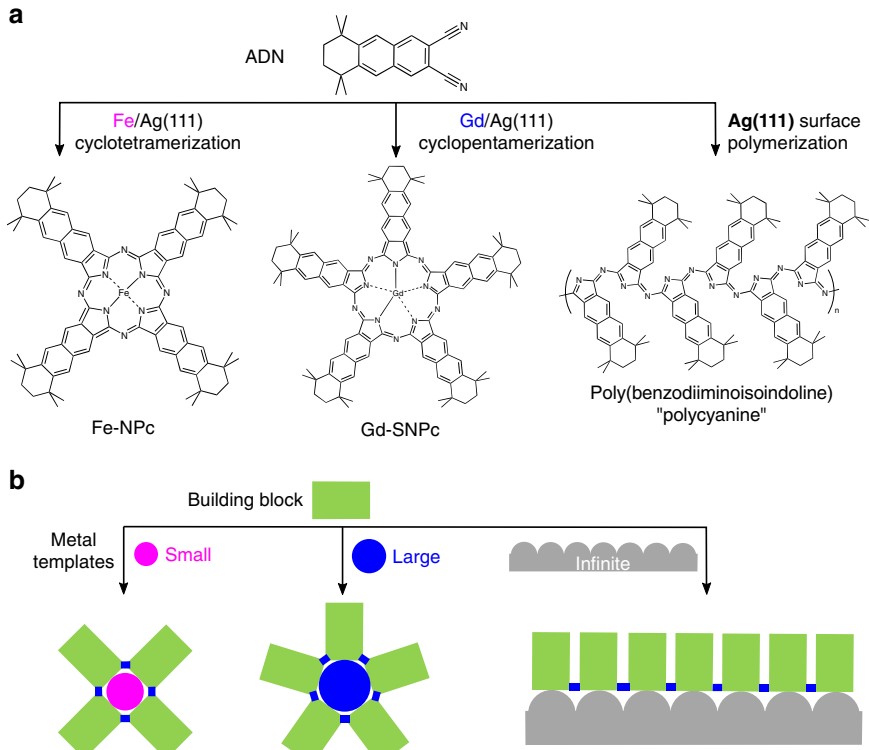

**Fig. 1** Reaction scheme and concept. **a** Reactions of the precursor monomer 5,5,8,8-tetramethyl-5,6,7,8-tetrahydroanthracene-2,3-dicarbonitrile (ADN) with different metal templates (Fe, Gd) on Ag(111), resulting in the formation of iron-naphthalocyanine (Fe-NPc) and gadolinium-supernaphthalocyanine (Gd-SNPc). Using the bare Ag(111) surface as a laterally extended template leads to surface-catalyzed polymerization and formation of poly (benzodiiminoisoindoline), here named polycyanine. **b** On-surface template approach: control over the topology of the reaction products is achieved by using differently sized metal templates in 2D confinement

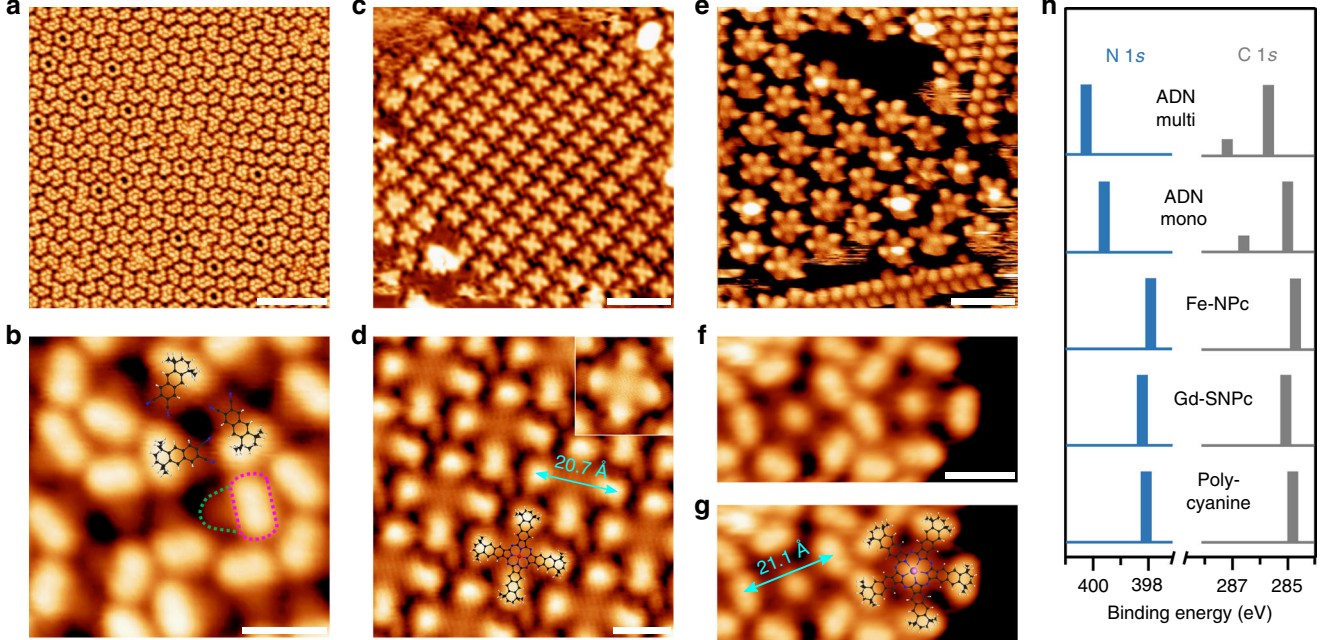

**Fig. 2** On-surface synthesis of Fe-NPc and Gd-SNPc. **a** STM image taken after vapor deposition of a monolayer of the precursor ADN onto Ag(111) at 300 K. **c, e** STM images taken after subsequent deposition of a low coverage of iron (0.13 ML) and gadolinium (0.06 ML), respectively, on the ADN monolayer followed by annealing to 450 K. **b, d, f** Magnified views of the samples in **a, c, e**, respectively. Molecular models of ADN and Fe-NPc are overlaid in **b, d**. The inset in **d** shows Fe-NPc imaged with the tunneling parameters $U = 1.2$ V, $I = 0.3$ nA. **g** shows **f** overlaid with a molecular model of Gd-SNPc. Black spheres represent carbon atoms; blue, nitrogen; white, hydrogen; red, iron; pink, gadolinium. Tunneling parameters: **a, b**, $U = -3.6$ V, $I = 0.08$ nA; **c**, $U = 1.2$ V, $I = 0.3$ nA; **d**, $U = 1.9$ V, $I = 0.13$ nA; **e**, $U = 0.76$ V, $I = 0.11$ nA; **f, g**, $U = -3.6$ V, $I = 0.11$ nA. Scale bars, **a**, 10 nm; **b, d, f, g**, 1.5 nm; **c**, 6 nm; **e**, 4 nm. **h** Binding energies of C 1s (gray) and N 1s (blue) XPS signals for ADN multi- and monolayers, Fe-NPc, Gd-SNPc, and polycyanine chains. The original XP spectra are shown in Supplementary Fig. 1

Fig. 1b: the small (Fe[40]), large (Gd[41–43]), and the infinite (silver surface) templates coordinate different numbers (four, five, and infinite) of building blocks that become eventually covalently connected. The benefits of the on-surface approach are apparent: the two-dimensional (2D) confinement posed by the surface allows only in-plane coordination of ligands with the metal. This results in the formation of the single-decker Gd-SNPc pentamer macrocycle rather than a bis(phthalocyaninato) double-decker complex. The perhaps most surprising advantage of this approach is that it enables the high-yield synthesis of isoindoline polymers, that is, polycyanine chains. The extended [-C = N-]$_n$ π-conjugated backbone of the polycyanine chain reported here makes it an intriguing model for an organic semiconducting polymer with an absorption band in the visible range.

## Results

**On-surface synthesis of Fe-NPc and Gd-SNPc**. The scanning tunneling microscope (STM) image in Fig. 2a shows a typical domain of an ADN monolayer (ML) on a Ag(111) surface. The magnified image in Fig. 2b reveals repeated structural motifs with an elongated bright protrusion (magenta dotted contour) and a weak tail (green dotted contour). Considering the chemical structure of ADN, the bright saddle and the weak tail are attributed to the tetramethyl-cyclohexane and the naphthalene dicarbonitrile moieties, respectively, as illustrated by the overlaid molecular models. The assignment to intact ADN is confirmed by the XP spectra taken on precisely the same sample as the STM images. The related X-ray photoelectron spectroscopy (XPS) binding energies (BEs) are summarized in Fig. 2h and the original spectra are shown in the Supplementary Fig. 1. The ML C 1s spectrum ("ADN mono") shows a major and a minor component

with BEs of 285.0 and 286.6 eV, respectively. Due to a higher electronegativity of nitrogen compared to hydrogen and carbon, the C 1s signal of the -C≡N group has a higher BE than that of the tetramethyl-tetrahydroanthracene backbone. Therefore, the major and minor components are related to the carbon atoms in tetramethyl-tetrahydroanthracene and carbonitrile (-C≡N) groups of ADN, respectively. The related N 1s spectrum shows a single peak at 399.6 eV, in agreement with the value of 399.5 eV reported for nitrogen in a -C≡N…H hydrogen bonding situation[44].

Vapor deposition of 0.13 ML iron atoms onto this ADN ML at 300 K, followed by annealing to 450 K, results in the formation of an iron-naphthalocyanine species with four tetramethyl-cyclohexane groups (Fe-NPc, Fig. 1a), as indicated by the cross-shaped features in Fig. 2c. The molecular structure has been confirmed by STM and XPS, as will be discussed in the following. First, the C 1s signal of the Fe-NPc sample in Fig. 2c shows only one peak at 284.7 eV (Fig. 2g and Supplementary Fig. 1). Compared to the spectrum of intact ADN, the disappearance of the -C≡N-related component supports the transformation from the -C≡N groups in ADN to the pyrrole and aza groups in Fe-NPc. Further evidence is provided by the related N 1s signal, which shifts towards lower BE by 1.7 eV upon formation of Fe-NPc. This result is consistent with the expected accumulation of negative charge at the N atoms when the dianionic naphthalocyanine ligand is formed. Based on the N 1s and C 1s spectra, it can be excluded that these cross-shaped objects are Fe(ADN)$_4$ coordination tetramers (Supplementary Fig. 2d) with four C≡N… Fe bonds, because four free -C≡N groups (along with four intact, but Fe-coordinated -C≡N groups) remain in a Fe(TCNB)$_4$ coordination tetramer. Second, the magnified view of the Fe-NPc layer (Fig. 1d and Supplementary Fig. 2e) reveals that each lobe of the cross-shaped species contains a bright protrusion at the

outer terminal resembling the appearance of the tetramethyl-cyclohexane group shown in Fig. 2b. In addition, the distance between two protrusions on opposite sides of the molecules is 20.7 ± 0.7 Å, in agreement with the expected value of 21.2 ± 0.2 Å (see Supplementary Fig. 2a–c)[45]. Notably, the Fe-NPc complexes in Fig. 2d do not show the Fe centers as protrusions, which is due to the high tip-sample bias of 1.9 V; reducing it to 1.2 V brightens the Fe center and the naphthalene moieties (inset in Fig. 2d).

In the formation of Fe-NPc, the iron atom acts both as a catalyst for the activation of -C≡N and as a template for coordinating the resulting isoindoline units. The small Fe atom has four in-plane coordination sites for -C≡N groups[40] and pyrrole ligands[40], which is consistent with the formation of a tetramer macrocycle. Using a much larger rare-earth metal, gadolinium, which allows for five in-plane coordination sites for -C≡N groups[43–45], enables the on-surface synthesis of the Gd-SNPc pentamer macrocycle, as indicated by the five-lobed pentamer structures in Fig. 2e. The sample in Fig. 2e was prepared by deposition of 0.06 ML Gd onto a complete ML of ADN, followed by annealing to 450 K. The structure of Gd-SNPc is corroborated by the following considerations: Fig. 2f shows the magnified STM image of the pentamer structure in Fig. 2e with high tunneling bias (−3.6 V, identical to that in Fig. 2b). It reveals that the pentamer consists of five axially arranged lobes surrounding a bright protrusion. The contour of each lobe resembles that of the tetramethyl-cyclohexane moiety of intact ADN (Fig. 2b). Therefore, the five lobes and the central bright protrusion are assigned to five tetramethyl-cyclohexane groups and the central Gd atom, respectively, as shown by the overlaid model in Fig. 2g. The distance between the two outermost lobes is 21.1 ± 0.5 Å (Fig. 2g), which agrees well with the expected distance of 21.2 ± 0.5 Å for Gd-SNPc (see Supplementary Fig. 4). Noteworthy, the different appearances of the five lobes of the pentamer structure in Fig. 2e, f arise from the different tunneling conditions. Similar to Fe-NPc (Fig. 2d, inset), imaging Gd-SNPc with a low positive bias of 0.76 V leads to the highlighting of the naphthalene backbone and the Gd metal center, as is illustrated by Fig. 2e. Conversely, a high negative bias (−3.6 V) results in the highlighting of the tetramethyl-cyclohexane moieties and the Gd center, as shown by Fig. 2f and Supplementary Fig. 5a (the latter presents a larger island of Gd-SNPc). Note that STM imaging is more difficult for Gd-SNPc than for Fe-NPc because the former has a five-fold symmetry. Since tiling of an Euclidian plane with regular pentagons is not possible, Gd-SNPc cannot form long-range ordered structures. The residual intermolecular space leads to increased lateral mobility and thus to a fuzzy appearance. The XPS data (Fig. 2h and Supplementary Fig. 1) prove the transformation of the -C≡N groups in ADN to the pyrrole and aza groups in Gd-SNPc with the same arguments as for Fe-NPc.

In order to compare the electronic properties between Fe-NPc and Gd-SNPc, we have calculated their frontier orbitals and the HOMO/LUMO (highest occupied molecular orbital/lowest unoccupied molecular orbital) gaps. The gap of Gd-SNPc is smaller by 0.32 eV than that of Fe-NPc, indicating a red-shifted absorption due to the extended π-conjugation pathway. As shown by Supplementary Figs. 17 and 18, the LUMOs of both Gd-SNPc and Fe-NPc are mixed ligand–metal orbitals. The SOMO of Gd-SNPc is mainly localized at the Gd(II) metal center, whereas the HOMO of Fe-NPc has predominant ligand character. Further details on the electronic structure of the two macrocycles can be found in the Supplementary Discussion.

**On-surface synthesis of polycyanine chains**. For both macro-cycles, the size of the π-conjugated system is determined by the sizes of the metal–atom templates. The further extension of the π-

system thus requires even larger templates (Fig. 1b). In the fol-lowing, we demonstrate that the Ag(111) surface itself acts as a quasi-infinitely extended template for the polymerization of ADN. A prerequisite is a high surface concentration of ADN, because it favors the growth of polymer chains, whereas a low surface concentration favors cyclization, according to the on-surface high-dilution principle[46,47]. Therefore, an ADN multi-layer was employed to maximize the concentration at the bottom-most layer on the Ag(111) surface during the reaction. The direct annealing of 5.8 layers of ADN (deposited at 300 K) to 450 K leads to the linear polymerization of ADN into polycyanine chains (Fig. 1a), as shown by the double-row-shaped chain structures in Fig. 3a. The excess ADN molecules desorb. The structure of this polycyanine chain was confirmed by morpho-logical and spectroscopic evidence as discussed below.

Figure 3b displays the magnified view of an isolated polycyanine chain, revealing a repeat unit with features resembling the ADN molecule in Fig. 2b. These repeat units are connected alternately at their tails, indicating bonding at the positions of the former -C≡N groups. The periodicity and the width, that is, center–center distances of the bright oval protrusions of the chain, are 8.8 ± 0.5 and 15.1 ± 0.5 Å, respec-tively. These values agree with the density functional theory (DFT) calculated periodicity (8.6 Å) and width (14.9 Å) of the polycyanine chains adsorbed on Ag(111) (Fig. 3c). We can exclude that the chains represent assemblies of intact ADN molecules connected through coordination or van der Waals bonds, because this would lead to a much larger periodicity. The absence of -C≡N groups in the chains is further confirmed by the C 1s and N 1s spectra taken on the sample in Fig. 3a (see Fig. 2h and Supplementary Fig. 1). The covalent linking between the chain elements is also supported by lateral manipulation experiments with the STM tip. As shown in Fig. 4, complete chains can be moved across the surface or be reversibly deformed without disintegration, revealing high mechanical stability.

The chains extend along the high-symmetry directions of the substrate and have a periodicity of three times the Ag(111) surface lattice constant, that is, they show point-on-line commensurability with respect to the substrate (Supplementary Fig. 6). The free-standing chain optimized with density functional theory (DFT) on the GGA level including dispersion correction (PBE-D3) shows a lattice mismatch of only 0.4%, underlining the commensurate character (Supplementary Table 2). There is good agreement between the experimental and DFT-simulated STM images (Fig. 3d).

## Discussion

Figure 5 shows the calculated electronic band structure of the chain and the related frontier crystal orbitals. Since the GGA approach is known to severely underestimate band gaps, a more reliable value of 2.0 eV is derived from range-separated hybrid density functional computations (HSE06)[48]. The calculated band gap corresponds to a visible-range absorption of ~600 nm. The effective mass of the lowest unoccupied crystal orbital (LUCO) is calculated to be 0.347, indicating that the free-standing poly-cyanine is a semiconducting polymer[49,50]. The LUCO is deloca-lized along the [-C=N-]$_n$ backbone and accordingly shows a large dispersion. Regarding the occupied bands, the two highest occupied crystal orbitals (HOCO, HOCO-1) are rather localized on the benzodiiminoisoindoline units and the N atoms, respec-tively. The lack of orbital overlap is in line with the low-dispersion nature of these bands. However, HOCO-2 and HOCO-3 show higher dispersion due to the moderate overlap of the non-bonding electron pairs at the nitrogen atoms.

There are two prerequisites for the on-surface formation of polycyanine chains. First, a high surface concentration of ADN

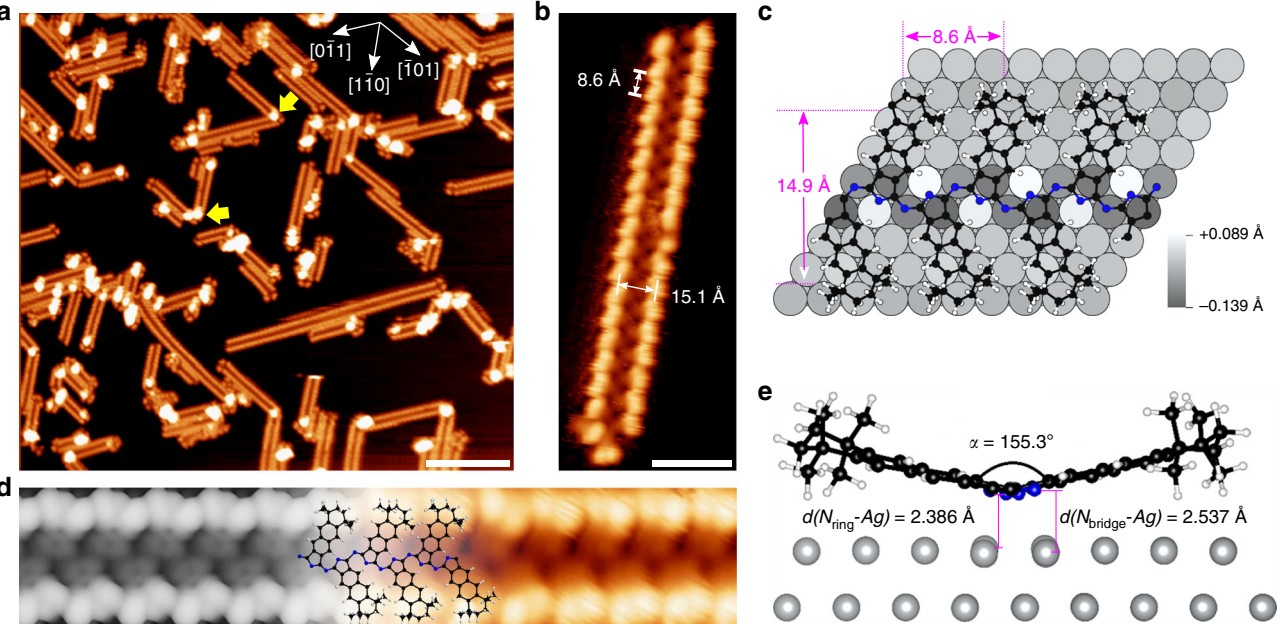

**Fig. 3** On-surface synthesis of polycyanine chains. **a** Overview STM image of the polycyanine chains, taken after deposition of 5.8 layers of ADN on Ag(111) at 300 K followed by annealing to 450 K. The white vectors indicate the three high-symmetry directions of the substrate. The yellow arrows indicate protrusions attributed to chain segments that are uplifted for steric reasons (see Supplementary Fig. 8 for details). **b** Magnified STM image of a single chain observed in the sample of **a**. **c**, **e** Top and side views of the DFT-calculated adsorption model of the polycyanine chain on a Ag(111) slab. The adsorbate-induced vertical displacement of the top-layer Ag atoms is illustrated with a gray scale in **c**. The distances of the N atoms in the pyrrole ring ($N_{ring}$) and the aza-bridging N atoms ($N_{bridge}$) with respect to the non-corrugated surface plane as well as the inter-plane angle $\alpha$ between the alternating benzodiiminoisoindoline moieties (see Supplementary Fig. 11 for a definition of $\alpha$) are shown in **e**. **d** Comparison of the DFT-calculated (left) and experimental (right) STM images of the polycyanine chain on Ag(111). A molecular model of the polycyanine chain is overlaid in the central part. Black spheres represent carbon atoms; blue, nitrogen; white, hydrogen; gray, silver. Tunneling parameters: **a**, $U = -3.6$ V, $I = 0.14$ nA; **b**, $U = 0.20$ V, $I = 0.10$ nA; **d**, $U = 0.55$ V, $I = 0.06$ nA. Scale bars, **a**, 15 nm; **b**, 3 nm

monomers is necessary, because otherwise cyclic tetramer macrocycles are formed (Supplementary Fig. 7). Since ADN partially desorbs at the reaction temperature of 450 K, ensuring a sufficient surface concentration during the reaction requires initial deposition of a multilayer. Second, the non-planar tetramethyl-cyclohexane group in ADN is probably critical for a successful reaction. As shown by the DFT-calculated adsorption model in Fig. 3e, the benzodiiminoisoindoline unit has a tilted adsorption configuration. Surprisingly, the DFT-optimized structure of the free-standing chain shows that this tilt is an intrinsic feature, since a butterfly angle of $\alpha = 140.9°$ (Supplementary Table 2) is calculated, which is close to that of the adsorbed chain (155.3°, Fig. 3e). The tilted conformation optimizes the conjugation between the non-bonding electron pairs at the nitrogen atoms and the $\pi$-electron system of the chain (see the Supplementary Figs. 12–16 and the related discussion in the Supplementary Discussion). Without the bulky tetramethyl-cyclohexane unit in the ADN precursor, the chain would be forced to assume a planar adsorption geometry, which is energetically unfavorable. The side groups in the ADN precursor therefore help the chain to maintain its optimal conformation on the surface. The adsorption energy for a unit cell containing two former ADN units is computed to $E_{ads} = -378$ kJ mol$^{-1}$, which mainly comes from dispersion interactions (Supplementary Fig. 11). Nevertheless, bonding analysis of a model system (Supplementary Fig. 10) with an energy decomposition analysis for extended systems[51] shows that the nitrogen atoms in the pyrrole rings ($N_{ring}$) show considerable attractive electrostatic and orbital interaction with the surface—an indicator for directional dative bonding (Supplementary Table 1).

Further evidence for the bonding of $N_{ring}$ to the Ag surface is provided by the adsorbate-induced vertical displacements of the

top-layer Ag atoms (Fig. 3c). Remarkably, the two sets of nitrogen atoms have opposite effects: the N atoms in the pyrrole rings ($N_{ring}$) lift up the first-layer Ag atoms, revealing attractive bonding Ag-$N_{ring}$ interactions. In contrast, the aza-bridging N atoms ($N_{bridge}$) push the Ag atoms into the surface, indicating repulsive interactions. The different bonding behavior of the two N species, which is confirmed by the different Ag-N distances shown in Fig. 3e, reveals an interesting analogy to the cyclic complexes, where only $N_{ring}$ binds to the metal, but not $N_{bridge}$. This analogy can be taken even further, if the surface-to-chain electron transfer is considered: per unit cell, the chain receives 0.58 electrons (see Supplementary Fig. 10) from the surface, just as the cyclic ligands receive electron density from their metal centers. There is, however, also an important difference: the cyclic ligands receive two electrons from the metal center to establish a Hückel-aromatic $18\pi$ electron conjugation path, that is, they are dianions (see Supplementary Fig. 9). In contrast, there is no such requirement regarding the electron count along the open-chain conjugation path. Thus, the chain can remain formally neutral and only fractional electron transfer between surface and molecule occurs.

We have achieved the template-controlled on-surface synthesis of an iron-naphthalocyanine, a gadolinium-supernaphthalocyanine, and a poly(benzodiiminoisoindoline) chain on a silver (111) surface. These syntheses start from an *ortho*-dicarbonitrile monomer (ADN) and employ template effects of iron atoms, gadolinium atoms, and the bare silver (111) surface, respectively. They also take advantage of the on-surface high-dilution principle. The different sizes of the metal templates enable the coordination and covalent linkage of different numbers of ADN monomers. STM and DFT investigations reveal that the intrinsically non-planar, semi-conducting poly(benzodiiminoisoindoline) chain is commensurate

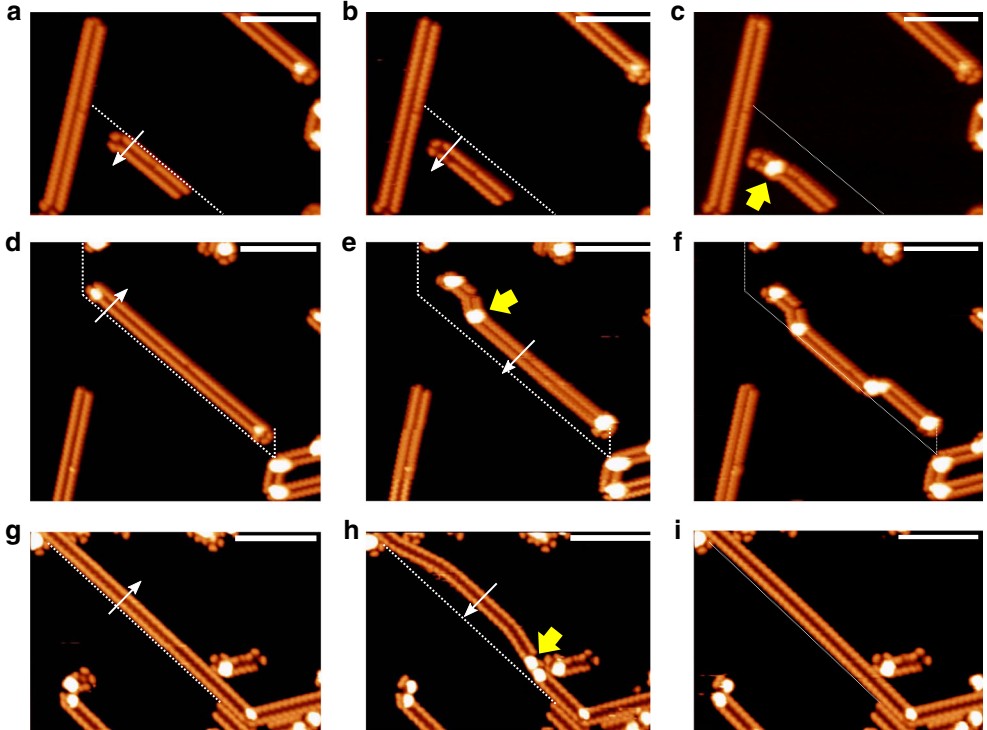

**Fig. 4** STM images of lateral manipulation steps of polycyanine chains. The manipulations were performed with the STM tip at the positions and along the directions of the white arrows. The initial locations of the chains are indicated by the dashed lines. **a**, **b** Manipulation of a short isolated chain moves the entire chain across the surface, while retaining its orientation. **b**, **c** Additional manipulation of this chain leads to further displacement and formation of a bent with a bright protrusion (yellow arrow in **c**, which is attributed to an uplifted chain segment (see Supplementary Fig. 8). These protrusions are induced by sterical hindrance and also occur at sharp bends, that is, when a chain changes its orientation, as marked by yellow arrows in Fig. 3a. **d–f** Manipulation of a longer chain, resulting in the formation of bends with related protrusions, but no rupture of the chain. **g–i** Reversible manipulation of a chain anchored at both ends. The induced protrusions appearing in **h** are reversed by pushing in the opposite direction (see **i**). Tunneling parameters: **a–f**, $U = -3.6$ V, $I = 0.09$ nA; **g–i**, $U = -3.7$ V, $I = 0.12$ nA. All the scale bars are 10 nm

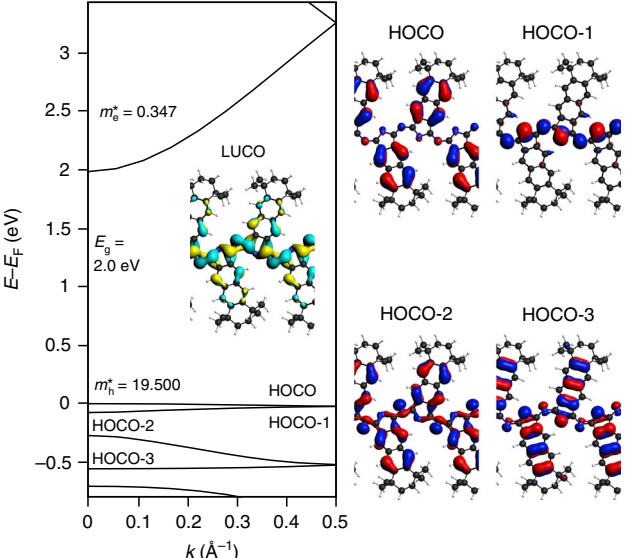

**Fig. 5** Band structure and band gap ($E_g$, HSE06) of polycyanine chains. The crystal orbitals at the Γ-point (LUCO/HOCO: lowest unoccupied / highest occupied crystal orbital) are shown together with effective masses for electron ($m_e^*$) and hole ($m_h^*$)

with the Ag surface, shows high mechanical stability and flexibility, and has a small band gap. The extended π-conjugated backbone makes it a potential candidate for applications in optoelectronics. Therefore, the protocols reported in this study may pave the way for

the discovery of multifunctional Pc-based molecular materials by control of the size of the π-conjugated system.

## Methods

**Synthesis**. 5,5,8,8-Tetramethyl-5,6,7,8-tetrahydroanthracene-2,3-dicarbonitrile (ADN) was prepared by using a published procedure as described in the Supplementary Discussion. The details of the synthesis and characterization of ADN are also included in the Supplementary Discussion.

**STM and XPS**. Theory: Density functional theory (DFT) calculations were performed using the GGA-type PBE functional for the structural optimization including a semiempirical dispersion-correction term (DFT-D3). The band structure was derived with the range-separated hybrid functional HSE06. STM simulations used the Tersoff-Hamann approach. Details are found in the Supplementary Discussion.

STM measurements were performed in an ultrahigh vacuum (UHV) system (base pressure $1 \times 10^{-10}$ mbar) equipped with a SPECS STM Aarhus 150 STM and a photoelectron spectrometer with SPECS Phoibos 150 electron energy analyzer. All bias voltages refer to the sample, and the images were recorded in constant current mode. Moderate filtering (Gaussian smooth, background subtraction) was applied for noise reduction. The Ag(111) single crystals with an alignment of better than 0.1° relative to the nominal orientation were purchased from MaTecK (Germany). Preparation of a clean and well-defined Ag(111) surface was achieved by cycles of bombardment with Ar$^+$ ions and annealing at 850 K. The ADN precursor was sublimed at 423 K in UHV from homebuilt Knudsen cell evaporators. Fe and Gd were deposited onto the ADN layer with a low flux of 0.04 nm/min, as measured with a quartz crystal microbalance. One monolayer of ADN, Fe, and Gd is defined as 0.047 ADN molecules, 1.31 Fe atoms, and 0.64 Gd atoms per surface Ag atom, respectively.

## Data availability

The data supporting the findings of this study are available from the corresponding authors upon request.

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

## Acknowledgements

Financial support by the Deutsche Forschungsgemeinschaft (DFG) through projects 223848855 - SFB 1083 and GO1812/2−1 is gratefully acknowledged. Q.F. thanks the Alexander von Humboldt-Foundation for a Research Fellowship for Postdoctoral Researchers. We thank the HRZ Marburg, CSC LOEWE Frankfurt and HLR Stuttgart for computational resources.

## Author contributions

J.M.G., Q.F., and J.S. conceived this project. Q.F., K.G., and M.Z. performed the STM/XPS experiments and analysis. M.L. and J.S. performed the multistep synthesis of the organic building block. R.T. and J.-N.L. performed the calculations. J.M.G. and Q.F. co-wrote the draft of this paper. All authors discussed the results and commented on the manuscript.

## Competing interests

The authors declare no competing interests.

**Additional information**

