## [Peer Review File · Nature Communications]

Reviewers' Comments:

Reviewer #1:

Remarks to the Author:

This paper provides a novel way to produce new super-phthalocyanine materials with interesting electronic and molecular structures. The data is of extremely high quality and the authors definitively prove the structures they propose for the cyano-based binding in these materials. The polymer is especially interesting as an optoelectronic material and would be much more so if it turns out that the process can be carried out with high turnover at the solution-solid interface.

One point has me quite curious. Why are the tetramethylcyclohexane groups so bright, even at 0.7 volts? How do they have any LDOS at 0.7 V relative to the naphthalocyanine? Is it possible that those groups are undergoing some form of cyceization on the Ag surface? Just being physically high doesn't seem like enough to outweigh the resonant tunneling through the naphthalocyanine at low bias.

Reviewer #2:

Remarks to the Author:

Gottfried and coworkers report an elegant control of on-surface reactivity, demonstrating formation of phthalocyanine (reported before for similar precursors) and novel superphthalocyanine and linear polyisoidoleimine, depending on the metal catalyst. Such diversity and selectivity of the covalent chemistry on the same surface (Ag(111)) is unprecedented. The polymer is of particular interest as a semiconductor and the results provide important insights in the structure of such pi-electron systems. The conclusions are fully supported by STM images and tip manipulation (of the polymer chain), complemented by thorough DFT analysis and XPS measurements. I think the work is of high novelty and significance and should be accepted in Nature Comm. However, the manuscript requires revisions/clarifications on the points below.

1) The Introduction provides a nice general summary on porphyrins and phthalocyanines. However, the specific prior work on on-surface phthalocyanine synthesis (while cited later in the text – Abel et al.) is not discussed. Likewise, no context to the prior on-surface macrocyclization/polymerization is given. I would argue that these present a more relevant prior knowledge and should be contextualized in the Introduction.

2) While highlighting synthetic limitations for superphthalocyanines, it seems relevant to note that boron subphthalocyanines are readily accessible and widely used.

3) The manuscript uses the term Q-band to describe the expected optical property of the polyisoidoleimine. It is a common notation differentiating the distinct S0-S1 and S0-S2 transitions of porphyrins and related macrocycles. Its use for conjugated polymers is unclear.

4) How were the reported uncertainties derived? Could the spacing between two blobs with irregular shape and ~10 Å size (eg, Fig. S2e) be defined with 0.3 Å uncertainty? What is the origin of uncertainty (0.2 Å) on DFT-calculated distances?

5) Related to the previous point, the statement "It can be excluded that these cross-shaped objects represent Fe(ADN)₄..." may need to be rephrased. The conclusion is well supported by XPS. As for the image appearance, there are substantial differences between the observed contrast and the Fe-NPC structure (Fig. S2), likely due to the substrate DOS; not clear if the deviations for the Fe(ADN)₄ structure are much larger.

6) The discussion of the band-gap may need to be rephrased. The PBE gap is discussed as underestimated but not actually provided (the reader has to go to SI). The word "improved" for the value of the gap is confusing. Besides, if the HSE and BB09 functionals "improve" the results, why only the PBE data for the band structure is given in the SI?

7) I think it is misleading to refer to 600 nm gap as an evidence that free-standing polycyanine is a

semiconducting polymer. The gap itself (providing it exists) has no bearing of the electrical conductivity. What matters is the energy of the individual bands and, more importantly, their width. In this context, the dispersion of the LUMO (but not HOMO) should be discussed. It suggests the polymer is good for electron, but not hole transport. I suggest calculating the electron effective mass in the LUMO.

8) The origin of the twisted geometry of the polymer is not really in the scope of this work. However, if the authors wish to discuss it, they need to make sure to bring clarity, not confusion. The twist can indeed increase the conjugation between the non-bonding electron pairs and the pi-electron system, but it also compromises the conjugation of the *bonding* pi-electron on that nitrogen. The latter is generally more important in conjugated azomethines; what makes it different here? I read carefully the explanation of the model studies in SI, but remain unconvinced. The attempt to address the question of electronic vs steric effect is well conceived but oversimplified. The two are obviously connected: the conjugation controls bond lengths and angles which in turn affects the sterics (eg, shorter C-N bonds increases steric transannular repulsion). What are the distances between the 'repulsive' atoms in the models M1-M8 (N...H, H...H)? Are the trends in the dihedral angle reproducible (at least qualitatively) with hybrid DFT? The unit cell is sufficiently small for higher-level ab initio calculations.

9) In drawing an analogy (last paragraph) of the Ag...N bonding in the Pc cycle and in the polymer, one may need to take into account that the stable form of the metal-free Pc is dianionic (or protonated), while the proposed structure of the polymer is neutral.

10) Analysis of the polymer/metal bonding is hard to follow. The Table S1 contains individual energy terms; none of them are defined (one could guess the meaning of some, but not all)

Dima Perepichka

Reviewer #3:

Remarks to the Author:

The work of Fan et al. describes the synthesis of a superphthalocyanine and a polycyanine on a silver surface. Gd atoms were used as catalysts to create Gd-pentamer macrocycles, whereas Fe only affords 'normal' tetrameric phthalocyanines. Through tuning the on-surface concentration, polycyanine chains can be polymerized, which withstand nano-manipulation on the molecular level as a consequence of the covalent bonds that stabilize the polymer chains. Scanning tunneling microscopy is used as a tool to track the reaction on the local level and the topography of precursor and products. X-ray photoelectron spectroscopy offers additional chemical insight and density functional theory puts the author's interpretations on firm ground.

The manuscript offers a sufficiently complete description with experimental and computational data, both thoroughly carried out. The use of a template for the synthesis of superphthalocyanines is novel, as is the synthesis of long polycyanine polymers, both very hard to synthesize with 'standard' synthetic chemistry methods.

The work is motivated by the authors by the design of materials with particular electronic structure such as red-shifted Q-bands. This theme is introduced in the abstract and picked up again in the introduction. A not very well placed section on page 7 is devoted to the theoretical exploration of the electronic structure of the polycyanine polymers (it tears apart the experimental results/discussion). For some reason, a discussion of the theoretical Gd-superphthalocyanine electronic structure is missing here, which is strange bearing in mind the motivation of the manuscript. How do HOMO/LUMO compare in the (super)phthalocyanines, e.g. what effect has the 5-fold symmetry on the frontier orbitals?

The manuscript would benefit greatly from experimental data that elucidate the electronic structure of

both superPc and polycyanine, possibly in the form of scanning tunneling spectroscopy or ultraviolet photoelectron spectroscopy.

Minor questions/remarks:

Was Gd measured in XPS? Can a shift in binding energy be observed before/after incorporation in the superphthalocyanine? This would give additional support to the author's hypotheses.

Fig. 3a shows many 1D polycyanine chains but also the occasional features that resemble the superphthalocyanine. What is the origin of these structures?

Reply to the reviewers' comments

We thank the reviewers for their competent and careful evaluation of our manuscript. Stimulated by the reviewers' comments, we have performed extensive additional work, especially DFT calculations, which further confirm our conclusions. On this basis, we have revised the manuscript and the supporting information and have added new material, including several figures. The most important additions include:

- (1) Extensive DFT calculations to further clarify the intrinsic butterfly-deformation of the polycyanine chain, including a Walsh diagram to identify the relevant orbitals (Figures S10-S15 in the SI).
- (2) Electronic structure calculations of Gd-SNPc and Fe-NPc, including the HOMO-LUMO gap and images of the frontier orbitals (Figures S16-S17 in the SI).
- (3) New STM images and height profiles to explain lateral distance measurements (Figure S3 in SI).
- (4) Aromaticity considerations for cyclic versus chain-like conjugation (Figure S18 in the SI).
- (5) A diagram with the band structure of the chain was moved to the manuscript, including pictures of the frontier crystal orbitals and the effective masses (new Figure 5).

Below, we reproduce the reviewer's comments together with our detailed responses. Our submission also includes a PDF file of the manuscript and the SI with highlighted changes.

Reviewer 1:

Comment: This paper provides a novel way to produce new super-phthalocyanine materials with interesting electronic and molecular structures. The data is of extremely high quality and the authors definitively prove the structures they propose for the cyano-based binding in these materials. The polymer is especially interesting as an optoelectronic material and would be much more so if it turns out that the process can be carried out with high turnover at the solution-solid interface.

Response: We thank the reviewer for the appreciation of our work and the helpful comments.

Comment: One point has me quite curious. Why are the tetramethylcyclohexane groups so bright, even at 0.7 volts? How do they have any LDOS at 0.7 V relative to the naphthalocyanine? Is it possible that those groups are undergoing some form of cyclization on the Ag surface? Just being physically high doesn't seem like enough to outweigh the resonant tunneling through the naphthalocyanine at low bias.

Response: We thank the reviewer for his constructive questions. Obviously, there is a misunderstanding regarding the apparent height of the tetramethylcyclohexane groups. The tetramethyl-cyclohexane groups are bright only when measured with a high bias (e.g., -3.6 V). They show very low LDOS at around 0.7 V, which leads to the absence of bright-oval feature of the tetramethylcyclohexane groups in both of the STM images of Fe-NPc (STM image in Fig. 2e) and Gd-SNPc (STM image in Fig. 2d) measured with a bias of 0.76 V. The lobes shown in these two STM images are therefore attributed to the planar naphthalene backbone. The related text about this point reads as follows: "Noteworthy, the different appearances of the five lobes of the pentamer structure in Figure 2e and 1f1 arise from the different tunneling conditions. Similar to Fe-NPc (Figure 2d, inset), imaging Gd-SNPc with a low positive bias of 0.76 V leads to the highlighting of the naphthalene backbone and the Gd metal center, as is illustrated by Figure 2e. Conversely, a high negative bias (-3.6

V) results in the highlighting of the tetramethyl-cyclohexane moieties and the Gd center, as shown by Figure 2f1 and Figure S4a (the latter shows a larger island of Gd-SNPc).”

The cyclisation of tetramethyl-cyclohexane groups on Ag(111) surface is unlikely. According to previous work, the C-H bond scission is reported to occur on the more reactive Cu(111) surface at temperatures higher than 470 K. [Treier, M. et al. Surface-assisted cyclodehydrogenation provides a synthetic route towards easily processable and chemically tailored nanographenes. *Nature Chemistry* 3, 61-67 (2011); Fan, Q. et al. Precise monoselective aromatic C–H bond activation by chemisorption of *meta*-aryne on a metal surface. *Journal of the American Chemical Society* 140, 7526-7532 (2018).] Therefore, on the less reactive Ag(111) surface, annealing the precursor molecules to 450 K is not sufficient to trigger cyclodehydrogenative coupling. This also agrees with previous work on porphyrins on Ag(111), where structural changes due to C-H bond scission require annealing to 550 K. [Di Santo, G. et al. Supramolecular Engineering through Temperature-Induced Chemical Modification of 2H-Tetraphenylporphyrin on Ag(111): Flat Phenyl Conformation and Possible Dehydrogenation Reactions. *Chem. Eur. J.* 17, 14354-14359 (2011); Papageorgiou, A. C. et al. Self-Terminating Protocol for an Interfacial Complexation Reaction in Vacuo by Metal–Organic Chemical Vapor Deposition. *ACS Nano* 7, 4520-4526 (2013)].

Reviewer 2:

Gottfried and coworkers report an elegant control of on-surface reactivity, demonstrating formation of phthalocyanine (reported before for similar precursors) and novel superphthalocyanine and linear polyiisindoleimine, depending on the metal catalyst. Such diversity and selectivity of the covalent chemistry on the same surface (Ag(111)) is unprecedented. The polymer is of particular interest as a semiconductor and the results provide important insights in the structure of such pi-electron systems. The conclusion are fully supported by STM images and tip manipulation (of the polymer chain), complemented by thorough DFT analysis and XPS measurements. I think the work is of high novelty and significance and should be accepted in Nature Comm. However, the manuscript requires revisions/clarifications on the points below.

Response: We appreciate the reviewer for the careful reading of our manuscript and the very insightful comments.

Comment: 1) The Introduction provides a nice general summary on porphyrins and phthalocyanines. However, the specific prior work on on-surface phthalocyanine synthesis (while cited later in the text – Abel et al.) is not discussed. Likewise, no context to the prior on-surface macrocyclization / polymerization is given. I would argue that these present a more relevant prior knowledge and should be contextualized in the Introduction.

Response: We thank the reviewer for pointing out this omission. The pioneering work regarding the cyclisation of dicarbonitriles should indeed be discussed earlier in the text. We therefore inserted the following text: “In recent years, it has been demonstrated that the co-deposition of *ortho*-dicarbonitriles with metal atoms (Mn [Koudia, M. & Abel, M. Step-by-step on-surface synthesis: From manganese phthalocyanines to their polymeric form. *Chem. Comm.* 50, 8565-8567 (2014); Piantek, M. et al. Manganese phthalocyanine derivatives synthesized by on-surface cyclotetramerization. *J. Phys. Chem. C* 118, 17895-17899 (2014).], Fe [Kezilebieke, S., Amokrane, A., Abel, M. & Bucher, J. P. Hierarchy of chemical bonding in the synthesis of Fe-phthalocyanine on metal surfaces: A local spectroscopy

approach. *J. Phys. Chem. Lett.* **5**, 3175-3182 (2014).]) on surfaces can result in macrocyclisation and formation of regular Pcs. Expanding on this approach, we report here ..."

Comment: 2) While highlighting synthetic limitations for superphthalocyanines, it seems relevant to note that boron subphthalocyanines are readily accessible and widely used.

Response: We thank the reviewer for his constructive suggestions. We included this aspect by making the following modification: "Therefore, changing the number of π -conjugated isoindoline units in phthalocyanines can produce molecular materials with properties that cannot be achieved by the corresponding expanded (or contracted) porphyrins. While the synthesis of *contracted* boron subphthalocyanines with three isoindoline units is well established, [Claessens, C. G., Gonzalez-Rodriguez, D., Rodriguez-Morgade, M. S., Medina, A. & Torres, T. Subphthalocyanines, subporphyrins, and subporphyrins: Singular nonplanar aromatic systems. *Chem. Rev.* **114**, 2192-2277 (2014).] the development of *expanded* Pcs is impeded by the limited synthetic possibilities"

Comment: 3) The manuscript uses the term Q-band to describe the expected optical property of the polyisoindoleimine. It is a common notation differentiating the distinct S0-S1 and S0-S2 transitions of porphyrins and related macrocycles. Its use for conjugated polymers is unclear.

Response: The reviewer is completely right that the term Q-band is not applicable to the open-chain product. Therefore, we rephrased the related text and now use the term "absorption band". The related discussion then reads as follows: "In contrast, the extended $[-C=N-]_n$ π -conjugated backbone of the polycyanine chain reported here makes it an intriguing model for an organic semiconducting polymer with an absorption band in the visible range... The calculated band gap corresponds to a visible-range absorption of approximately 600 nm."

Comment: 4) How were the reported uncertainties derived? Could the spacing between two blobs with irregular shape and ~ 10 Å size (eg, Fig. S2e) be defined with 0.3 Å uncertainty? What is the origin of uncertainty (0.2 Å) on DFT-calculated distances?

Response: The uncertainties are derived from the variation of the distances by measuring several identical species. For instance, for the Fe-NPc, we measured the distances between the bright lobes of four cross-shaped species in the same STM images [as illustrated by Figure R1 below]. The difference of the distances may originate from the thermo-drift during the STM imaging, which causes different extent of deformation at different parts of the image.

The uncertainty of 0.3 Å in Fig. S2e is derived as show by Figure R1 as discussed in the following. Fig. R1 shows the apparent height profiles along the green lines overlaid on the STM images. The lobe-to-lobe distances (20.72 Å, 20.45 Å, 20.95 Å, and 20.62 Å) of four cross-shaped species, which are located at different parts of the image, have been measured. Therefore, the average value of these distances is 20.685 Å (≈ 20.7 Å), and the error bar is defined here as the largest deviation 20.95 – 20.685 Å = 0.265 Å (≈ 0.3 Å). To elucidate this aspects, we have added the derivation details of the uncertainties of the distances into supporting information (see Figure S3 and the corresponding discussions).

For the uncertainty of DFT-calculated distances, there might be a misunderstanding. We did not do the DFT calculations of the distances between the lobes of Fe-NPc and Gd-SNPc. Only the periodicity of the polyisoindoline chains has been calculated with DFT. The expected lobe-to-lobe distances of Fe-

NPc and Gd-SNPc is derived also from experimental data reported in previous work. The uncertainties of the distances therefore originates from the non-contact AFM image of the NPc in Fig. S2a, which are derived in a similar way to that in Fig. S2e.

Fig. R1. (a-d) Apparent height (right part) profiles along the green lines overlaid on different cross-shaped species in the STM images (left part). The three blue dotted lines are given as reference to find the full width at half maximum (FWHM) (the center dotted lines). The peak position is defined as the center of the FWHM.

Comment: 5) Related to the previous point, the statement “It can be excluded that these cross-shaped objects represent Fe(ADN)₄...” may need to be rephrased. The conclusion is well supported by XPS. As for the image appearance, there are substantial differences between the observed contrast and the Fe-NPc structure (Fig. S2), likely due to the substrate DOS; not clear if the deviations for the Fe(ADN)₄ structure are much larger.

Response: We thank the reviewer for his constructive recommendations. Indeed, the C 1s and N 1s spectra already well supported the transformation of the carbonitrile groups. So we revised the related section, which now reads as follows: "First, the C 1s signal of the Fe-NPc sample in Figure 2c shows only one peak at 284.7 eV (Figures 2g and S1). Compared to the spectrum of intact ADN, the disappearance of the -C≡N related component supports the transformation from the -C≡N groups in ADN to the pyrrole and aza groups in Fe-NPc. Further evidence is provided by the related N 1s signal, which shifts towards lower BE by 1.7 eV upon formation of Fe-NPc. This result is consistent with the expected accumulation of negative charge at the N atoms when the dianionic naphthalocyanine ligand is formed. Based on the N 1s and C 1s spectra, it can be excluded that these cross-shaped objects are Fe(ADN)₄ coordination tetramers (Figure S2d) with four C≡N⋯Fe bonds, because four free -C≡N groups (along with four intact, but Fe-coordinated -C≡N groups) remain in a Fe(TCNB)₄ coordination tetramer. Second, the magnified view of the Fe-NPc layer (Figures 1d and S2e) reveals that each lobe

of the cross-shaped species contains a bright protrusion at the outer terminal resembling the appearance of the tetramethyl-cyclohexane group shown in Figure 2b. In addition, the distance between two protrusions on opposite sides of the molecules is $20.7 \pm 0.3 \text{ \AA}$, in agreement with the expected value of $21.2 \pm 0.2 \text{ \AA}$ (see Figure S2a-c). [Mohn, F., Gross, L., Moll, N. & Meyer, G. Imaging the charge distribution within a single molecule. *Nat. Nanotechnol.* **7**, 227-231 (2012).] Notably, the Fe-NPc complexes in Figure 2d do not show the Fe centers as protrusions, which is due to the high tip-sample bias of 1.9 V; reducing it to 1.2 V brightens the Fe center and the naphthalene moieties (inset in Figure 2d)."

Comment: 6) The discussion of the band-gap may need to be rephrased. The PBE gap is discussed as underestimated but not actually provided (the reader has to go to SI). The word 'improved' for the value of the gap is confusing. Besides, if the HSE and BB09 functionals 'improve' the results, why only the PBE data for the band structure is given in the SI?

Response: We agree with the reviewer that showing the best available data is the approach to follow. We have recalculated the band structure with the range-separated hybrid functional HSE06 and added this into the main text as shown by the new Figure 5 (also Figure R2 shown below). We also became aware of potential issues with the TB09 functional regarding band gaps in 2D systems and decided to remove the value from manuscript and supporting information as it also does not add insight beyond the results from HSE06 which are now most prominently displayed. With these new calculation results, we revised the discussions related to the band-gap as follows: "Figure 5 shows the DFT-calculated electronic band structure (HSE06) of the chain and the related frontier crystal orbitals. Since PBE is known to underestimate band gaps, a more reliable value of 2.0 eV is derived at the hybrid density functional level (HSE06)."

Figure R2. Band structure and band gap (E_g , HSE06) of polycyanine chains. The crystal orbitals at the Γ -point (LUCO/HOCO: lowest unoccupied / highest occupied crystal orbital) are shown together with effective masses for electron (m_e^*) and hole (m_h^*).

Comment: 7) I think it is misleading to refer to 600 nm gap as an evidence that free-standing polycyanine is a semiconducting polymer. The gap itself (providing it exists) has no bearing of the electrical conductivity. What matters is the energy of the individual bands and, more importantly, their width. In this context, the dispersion of the LUMO (but not HOMO) should be discussed. It suggests the polymer is good for electron, but not hole transport. I suggest calculating the electron effective mass in the LUMO.

Response: We agree with the reviewer that 600 nm gap does not necessarily indicate the semiconductivity of the polymer. As suggested by the reviewer, we have calculated the electron and hole effective masses with a quadratic fit procedure as shown by Figure 5 (also Figure R2). The effective mass of the LUCO is $0.347 m_0$, which indicates the conducting properties of the polymer. The effective mass of $0.347 m_0$ is well within the range discussed in previous work (DOI: 10.1039/c6cp06101j, DOI: 10.1103/PhysRevB.68.035204). Therefore, we revised the discussion regarding the semiconductivity of the polymers as follows: “The calculated band gap corresponds to a visible-range absorption band of approximately 600 nm. The effective mass of the LUCO is calculated to be $0.347 m_0$, indicating that the free-standing polycyanine is a semiconducting polymer. [Gutzler, R. Band-structure engineering in conjugated 2D polymers. *Phys. Chem. Chem. Phys.* **18**, 29092-29100 (2016); Hutchison, G. R. *et al.* Electronic structure of conducting polymers: Limitations of oligomer extrapolation approximations and effects of heteroatoms. *Phys. Rev. B* **68**, 035204 (2003).]”

Comment: 8) The origin of the twisted geometry of the polymer is not really in the scope of this work. However, if the authors wish to discuss it, they need to make sure to bring clarity, not confusion. The twist can indeed increase the conjugation between the non-bonding electron pairs and the pi-electron system, but it also compromises the conjugation of the *bonding* pi-electron on that nitrogen. The latter is generally more important in conjugated azomethines; what makes it different here? I read carefully the explanation of the model studies in SI, but remain unconvinced. The attempt to address the question of electronic vs steric effect is well conceived but oversimplified. The two are obviously connected: the conjugation controls bond lengths and angles which in turn affects the sterics (eg, shorter C-N bonds increases steric transannular repulsion). What are the distances between the 'repulsive' atoms in the models M1-M8 (N...H, H...H)? Are the trends in the dihedral angle reproducible (at least qualitatively) with hybrid DFT? The unit cell is sufficiently small for higher-level ab initio calculations.

Response: We are happy to see the reviewer being interested in this interesting side aspect of the study. We reproduced the GGA results with a range-separated hybrid functional (HSE06) and see the same trends (see the comparison in Figure S11 and S12, also in Figure R3 and R4). Higher level optimization for the molecular model system (MP2) did not show significant deviations in the structure. Additionally, we now include a Walsh diagram of further simplified model system M9 in Figure S15 (see also Figure R5) showing the orbital shape and energies for different dihedral angles. It can be clearly seen that upon increasing the dihedral angle from 0° to the structural minimum of 46° the HOMO-3 representing non-bonding electron pair conjugation is strongly stabilized while the conjugation (orbitals HOMO, HOMO-1, HOMO-4) is decreased by a similar but slightly smaller amount. We also clarified the definition of the angle α to make it consistent with the bending values for the model systems. This supports our previous argumentation.

Therefore, we added the models calculated with higher-level HSE06 in the supporting information for comparison as shown by Figure S11 and S12. The corresponding discussion of this aspect has been added reading as follows: “Even for a very small model system (M9, Figure S13), the bending is found. In the case of this model, we use the dihedral angle $\theta < (\text{N-C-N-C})$ to describe the bending since a second

ring is not available for the definition of α in the same way as done in the original system. For this model system, we can set up a Walsh diagram (Figure S15) showing the orbital shape and energies for different dihedral angles θ . It can be clearly seen that upon increasing the dihedral angle from 0° to the structural minimum of 46° the HOMO-3 representing non-bonding electron pair conjugation is strongly stabilized while the conjugation (orbitals HOMO, HOMO-1, HOMO-4) is decreased by a similar but slightly smaller amount. This intrinsic bending thus stems mostly from the increased conjugation of the non-bonding electron pair at nitrogen.

Noteworthy, to achieve more accuracy of the models in Figure S11 and Figure S12, we reproduced the GGA results with a range-separated hybrid functional (HSE06) and see the same trends (see the comparisons in Figure S11 and S12). Higher level optimization for the molecular model system (MP2) did not show significant deviations in the structure."

Figure R3. Series of model systems for the identification of the origin of the out-of-plane bending (angle defined according to Figure S10). (M1) Simplified version of the polycyanine chain. (M2) H saturation

of half of the N atoms in the pyrrole rings (N_{ring}). (M3) H saturation of all N_{ring} . (M4) H saturation of all N_{ring} and isoelectronic exchange of a peripheral $-\text{CH}$ with N. (M5) Isoelectronic exchange of all N with $-\text{CH}$. (M6) H saturation of the peripheral $\text{C}=\text{C}$. (M7) H saturation of all bridging imines (N_{bridge}). (M8) Isoelectronic exchange of a peripheral $-\text{CH}$ with N. The calculated models in the left and right columns are obtained by PBE and HSE06 functionals, respectively.

Figure R4. A larger model system based on M7, but with an additional annulated benzene ring is planar, excluding steric repulsion (e.g. between the H atom at the benzene ring and the lone pair at the N atom in the pyrrole ring) as the cause for the tilted chain. The results at the top and bottom rows are calculated with PBE and HSE06 functionals, respectively.

Figure R5. Walsh diagram showing the dependence of the canonical Kohn-Sham orbital energies as function of the dihedral angle θ in the model system M9.

Comment: 9) In drawing an analogy (last paragraph) of the Ag...N bonding in the Pc cycle and in the polymer, one may need to take into account that the stable form of the metal-free Pc is dianionic (or protonated), while the proposed structure of the polymer is neutral.

Response: We thank the reviewer for drawing our attention to this point, at which our manuscript may be misleading. Indeed, the Pc ligand must be dianionic to gain its Hückel-aromatic 18π electron conjugation path. In contrast, there is no such requirement regarding the electron count along the open-chain conjugation path and thus the chain can remain formally neutral. To account for this aspect, we added the following text at the end of the main part of the manuscript (directly before the conclusion section): "There is, however, also an important difference: The cyclic ligands receive two electrons from the metal center to establish a Hückel-aromatic 18π electron conjugation path, i.e., they are dianions (see Figure S18). In contrast, there is no such requirement regarding the electron count along the open-chain conjugation path. Thus, the chain can remain formally neutral and only fractional electron transfer between surface and molecule occurs."

To further illustrate this point, we added a new figure to the SI:

Figure R6. (a)-(c) Conjugated tetraazaporphyrin macrocycles as models for the phthalocyanine macrocycles. The cyclic systems are stabilized by a Hückel-aromatic 18π electron conjugation path when they contain two NH hydrogens like in the free base (a) or are dianions (b), which occur in metal complexes. If neither condition is fulfilled, an anti-aromatic 16π conjugation path results, as shown in (c). In contrast, conjugation along the open-chain path (d) there does not pose such requirements regarding the electron count. The chain is shown in the quasi-cyclic conformation to illustrate the analogy to the anti-aromatic cyclic system in (c).

Comment: 10) Analysis of the polymer/metal bonding is hard to follow. The Table S1 contains individual energy terms; none of them are defined (one could guess the meaning of some, but not all)

Response: We thank the reviewer for his pointing out of this aspects. Now we added the definition of all the energy terms in the Supporting information, which makes it more readable. The description reads as follows: "Bonding analysis was performed at PBE-D3(BJ)/DZP with different k space samplings, using pEDA. The pEDA method allows to dissect the interaction energy between two fragments (here: molecule and surface) into well-defined quantities that allow to interpret the bonding in a system in a chemically meaningful way. The interaction energy (ΔE_{int}) is first divided into a dispersion term (ΔE_{disp}) and an electronic term (ΔE_{elec}). The actual pEDA procedure then decomposes ΔE_{elec} into contributions from Pauli repulsion (ΔE_{Pauli}), electrostatics (ΔE_{elstat}) and orbital interaction (ΔE_{orb}):

$$\Delta E_{\text{elec}} = \Delta E_{\text{Pauli}} + \Delta E_{\text{elstat}} + \Delta E_{\text{orb}}$$

This enables a quantitative analysis of the surface-adsorbate bonding."

Reviewer 3:

Comment: The work of Fan et al. describes the synthesis of a superphthalocyanine and a polycyanine on a silver surface. Gd atoms were used as catalysts to create Gd-pentamer macrocycles, whereas Fe only affords 'normal' tetrameric phthalocyanines. Through tuning the on-surface concentration, polycyanine chains can be polymerized, which withstand nano-manipulation on the molecular level as a consequence of the covalent bonds that stabilize the polymer chains. Scanning tunneling microscopy is used as a tool to track the reaction on the local level and the topography of precursor and products. X-ray photoelectron spectroscopy offers additional chemical insight and density functional theory puts the author's interpretations on firm ground.

The manuscript offers a sufficiently complete description with experimental and computational data, both thoroughly carried out. The use of a template for the synthesis of superphthalocyanines is novel, as is the synthesis of long polycyanine polymers, both very hard to synthesize with 'standard' synthetic chemistry methods.

the work is motivated by the authors by the design of materials with particular electronic structure such as red-shifted Q-bands. This theme is introduced in the abstract and picked up again in the introduction. A not very well placed section on page 7 is devoted to the theoretical exploration of the electronic structure of the polycyanine polymers (it tears apart the experimental results/discussion). For some reason, a discussion of the theoretical Gd-superphthalocyanine electronic structure is missing here, which is strange bearing in mind the motivation of the manuscript. How do HOMO/LUMO compare in the (super)phthalocyanines, e.g. what effect has the 5-fold symmetry on the frontier orbitals?

Response: We thank the reviewer for his constructive suggestions. We have done additional calculations regarding the HOMO/LUMO gap of the Gd-SNPc and Fe-NPc as shown by Figure S16 and S17 (also shown as Figure R7 and R8 below). The former is smaller than the latter by 0.32 eV. In both cases, the LUMO is a mixed ligand-metal orbital. The HOMO of Gd-SNPc shows mainly metal orbital (d_z^2) character while it is a ligand-based orbital for Fe-NPc. The results have to be taken as preliminary due to the demanding electronic structure of these large metal complexes. With these DFT-calculated results, we added the discussion regarding the comparison of the electronic properties between Gd-SNPc and Fe-NPc reading as follows: "In order to compare the electronic properties between Fe-NPc and Gd-SNPc, we have calculated their frontier orbitals and the HOMO/LUMO gaps. The gap of Gd-SNPc is smaller by 0.32 eV than that of Fe-NPc, indicating a red-shifted absorption due to the extended π -conjugation pathway. As shown by Figures S16 and S17, the LUMOs of both Gd-SNPc and Fe-NPc are mixed ligand-metal orbitals. The SOMO of Gd-SNPc is mainly localized at the Gd(II) metal center, whereas the HOMO of Fe-NPc has predominant ligand character. Further details on the electronic structure of the two macrocycles can be found in the Supplementary Information."

Additional information in the SI:

Figure R7. Singly-occupied molecular orbital (SOMO, left) and LUMO (right) of the Gd-SNPc as calculated with PBE0. The orbital energy gap amounts to 1.39 eV. Preliminary screening of electronic states with PBE identified eight unpaired electrons as the most favorable state. Seven of the unpaired electrons are located in low-lying 4f orbitals, here with α -spin. The eighth unpaired electron, of β -spin, is located in the d_{z^2} SOMO which would favor Gd(II) in the gas phase complex (an all- α configuration could not be converged). These findings indicate a complex electronic structure for Gd-SNPc which is likely not well reproduced with single-determinant methods. The molecule is nevertheless too large for more accurate wavefunction based multireference methods at the current stage. Additionally, on the surface the electronic state is expected to be different.

Figure R8. HOMO (left) and LUMO (right) of the Fe-NPc as calculated with PBE0. The orbital energy gap amounts to 1.71 eV. Of the electronic states investigated, a triplet state was the most favorable in accordance with Fe-PC literature [<http://dx.doi.org/10.1063/1.3554212>].

Comment: The manuscript would benefit greatly from experimental data that elucidate the electronic structure of both superPc and polycyanine, possibly in the form of scanning tunneling spectroscopy or ultraviolet photoelectron spectroscopy.

Response: We thank the reviewer for this kind suggestion. While the focus of our study is on the synthesis, measurements of the electronic properties using scanning tunneling spectroscopy (STS) or ultraviolet photoelectron spectroscopy would of course be interesting. Currently, we cannot perform reliable STS measurement with our variable-temperature STM because of instrumental limitations. For ultraviolet photoelectron spectroscopy, a very pure and laterally homogenous phase of the material is required. However, minor byproduct typically exists during the major formation of these two materials. Another challenge is the influence of the metal substrate on the electronic properties of these materials. Therefore, we are planning future collaborative project, in which we try to isolate these materials from the surface and measure their electronic properties without the influence of the metal surface. However, these studies are outside the scope of the current work.

Comment: Minor questions/remarks: Was Gd measured in XPS? Can a shift in binding energy be observed before/after incorporation in the superphthalocyanine? This would give additional support to the author's hypotheses.

Response: We thank the reviewer for giving us the opportunity to expand on this important point. Although the metal atom gets formally oxidized to its dication when the complex is formed, we do not necessarily expect a related binding energy (BE) shift for the adsorbed species. This is because of electron transfer from the Ag(111) surface. For FePc on Ag(111) [Bai, Y. *et al.* Direct Metalation of a Phthalocyanine Monolayer on Ag(111) with Coadsorbed Iron Atoms. *J. Phys. Chem. C* **112**, 6087-6092 (2008)], but also for the related Fe porphyrin complexes on Ag(111) [e.g., Hieringer, W. *et al.* The Surface Trans Effect: Influence of Axial Ligands on the Surface Chemical Bonds of Adsorbed Metalloporphyrins. *J. Am. Chem. Soc.* **133**, 6206-6222, (2011).; more examples in Gottfried, J. M. Surface Chemistry of Porphyrins and Phthalocyanines. *Surf. Sci. Rep.* **70**, 259-379 (2015).], the main M 2p signal appears at a M(0) position, i.e., there is no chemical shift between the metal before and after the complex formation. Similar findings were reported for the related Co porphyrin and phthalocyanine complexes [e.g., Flechtner, K., Kretschmann, A., Steinruck, H. P. & Gottfried, J. M. NO-induced reversible switching of the electronic interaction between a porphyrin-coordinated cobalt ion and a silver surface. *J. Am. Chem. Soc.* **129**, 12110 (2007)., more examples in Gottfried, J. M. Surface Chemistry of Porphyrins and Phthalocyanines. *Surf. Sci. Rep.* **70**, 259-379 (2015)]. The reason lies in the interaction of the semi- or unoccupied d_{z^2} orbital (or other d-orbitals) with the Ag(111) surface and electron transfer from the surface to the metal center. Since Gd in the complex has a semi-occupied d_{z^2} orbital, we expect a very similar behavior as in the related Co complexes, where the main Co 2p core level signal of the adsorbed complex on Ag(111) is at a Co(0) position. Therefore, such measurements are probably not suitable for supporting our case. This is why we focused on the N 1s and C 1s signals in this work. Nevertheless, we also measured Gd 3d and Gd 4d spectra. Initially, we used the Gd 3d signal, because it has the highest photoemission cross section at the given photon energy. However, it turned out that the Gd 3d signal is affected by a superimposed MNV Auger signal of Ag surface, which made it very difficult to differentiate the peak changes of Gd 3d. This holds especially as the amount of Gd atom in the on-surface reaction is very low (few percent of a monolayer). Therefore, we changed to measuring the Gd 4d instead, which is located above a relatively flat background, but has even lower intensity. The direct XPS measurement of the Gd 4d spectrum of 0.06 ML Gd (the amount was used to synthesize the phases for the STM measurements) shows almost no intensity of the Gd 4d peak even for very extended data collection times (several hours). Deposition of 0.64 ML (≈ 10 times the previous amount) Gd onto a monolayer of intact ADN molecules then results

in the spectra shown by Figure R9. The blue curve shows the fitted Gd 4d spectrum taken after direct deposition of 0.64 ML Gd onto ADN monolayer held at 300 K. Here, we expect the signal of Gd(0). The red curve shows the fitted Gd 4d spectrum taken after annealing 0.64 ML Gd/ADN to 450 K, where we expect the signal of the Gd(II). As can be seen, for the red curve, a tiny shoulder (green arrow) appears at a BE of ≈ 144 eV (compared ≈ 142 eV for the main peak of Gd(0)). This shoulder may originate from the Gd²⁺ in the formed Gd-SNPc. We do not expect that all Gd atoms to react, because a complete monolayer of Gd-SNPc contains less than 0.03 ML Gd, while we deposited 0.64 ML for the XPS experiment because of the very weak signal. This means that less than 5% of the total deposited Gd has reacted, while more than 95% remain metallic.

To summarize, we do not necessarily expect a reaction-related chemical shift, based on (our) previous XPS work with other metal tetrapyrrole complexes on Ag(111), due to surface interaction. The Gd 3d signal is obscured by a Ag Auger transition, and the Gd 4d signal has a low intensity. Therefore, reaction with the stoichiometric amount of Gd (<3% of a monolayer) cannot be studied with XPS. A 20-fold excess of Gd leads to a measurable Gd 4d signal (Figure R9), but reaction-related changes are necessarily small (<5% of the total signal). Although small changes occur, they cannot reliably be attributed to the reaction. Therefore, XPS measurements on Gd core levels are not suitable for studying the formation of Gd-SNPc under the given circumstances. The shown N 1s and C 1s spectra provide much better information.

Figure R9. The blue and red curves show the fitted Gd 4d spectra obtained after direct deposition of 0.64 ML Gd onto ADN monolayer held at 300 K (blue) and followed by annealing to 450 K (red).

Comment: Fig. 3a shows many 1D polycyanine chains but also the occasional features that resemble the superphthalocyanine. What is the origin of these structures?

Response: We thank the reviewer for the careful inspection of our data. Indeed, there are occasionally cyclic oligomers even under chain-growth conditions (i.e., high precursor concentrations). It must be noted that there is always a competition between ring and chain formation. Although chains form preferentially under high concentration conditions, rings still can occasionally form with a certain (low) probability, perhaps even without participation of a templating metal center. Therefore, these five-lobed features might be Ag_xSNPc species. Trapping of the Ag atom(s) may happen during or after formation of the ring.

Such five-lobed features (pointed by white arrows in Figure R10) have also been observed as minor species on the same sample as in Figure 2c. This might be also due to the occasional formation of pentamer, which subsequently captures one or two Fe atoms. Even larger cyclic hexamer and heptamer have occasionally been observed on this sample (Figure R10). Since we cannot elucidate these minority species unambiguously with the current techniques, we therefore think it is better to focus on the clearer majority species and study the side products in the future work.

Figure R10. Overview STM image of the same sample as in Figure 2c showing minority species with pentamer (pointed by white arrows), hexamer, and heptamer features.

Reviewers' Comments:

Reviewer #2:

Remarks to the Author:

The authors have done an excellent job at responding to all my (and other referees) concerns. I recommend accepting the paper. The authors may consider the below but no further review is needed.

Related to the authors response to my comment #4. I still think that rounding the uncertainty to 0.5Å would be less controversial. I understand that the statistical analysis of 4 points gives 0.3Å, but one should also consider the uncertainty of the individual measurements. Looking at the scale of Fig. S3 (the smallest unit is 2.5Å), it appears that the determining the maximum of a given peak already comes with a 0.5Å uncertainty. Also, I am still unclear as to the origin of the cited 0.2Å uncertainty of the "expected value":

"In addition, the distance between two protrusions on opposite sides of the molecules is $20.7 \pm 0.3 \text{ \AA}$, in agreement with the expected value of $21.2 \pm 0.2 \text{ \AA}$ (see Figure S2a-c)."

Dmitrii Perepichka

Reviewer #3:

Remarks to the Author:

The authors took great care and effort in answering the comments/questions of all three referees. The improved manuscript should be accepted for publication in Nature Communications.

Final revisions for manuscript NCOMMS-19-11643A

Response to the reviewers' comments

REVIEWERS' COMMENTS:

Reviewer #2 (Remarks to the Author):

The authors have done an excellent job at responding to all my (and other referees) concerns. I recommend accepting the paper. The authors may consider the below but no further review is needed. Related to the authors response to my comment #4. I still think that rounding the uncertainty to 0.5A would be less controversial. I understand that the statistical analysis of 4 points gives 0.3A, but one should also consider the uncertainty of the individual measurements.

Looking at the scale of Fig. S3 (the smallest unit is 2.5A), it appears that the determining the maximum of a given peak already comes with a 0.5A uncertainty. Also, I am still unclear as to the origin of the cited 0.2A uncertainty of the "expected value":

"In addition, the distance between two protrusions on opposite sides of the molecules is $20.7 \pm 0.3 \text{ \AA}$, in agreement with the expected value of $21.2 \pm 0.2 \text{ \AA}$ (see Figure S2a-c)."

Dmitrii Perepichka

Response: We thank the reviewer for insisting on this important point. We agree that the lateral distance uncertainty was underestimated. An important factor is the pixel resolution. The STM image in the Supplementary Figure 3 contains 512×512 pixels at a size of 10×10 nm², which means that one pixel corresponds already to a distance of approximately 0.2 Å (100 Å/512). An uncertainty of 0.5 Å, as suggested by the reviewer, is therefore more realistic. The same value is now used for the value obtained from literature AFM data. Originally, we assumed here a smaller uncertainty because of the higher pixel resolution of the AFM image, but considering other sources of uncertainty makes the higher value appear more realistic here as well.

Reviewer #3 (Remarks to the Author):

The authors took great care and effort in answering the comments/questions of all three referees. The improved manuscript should be accepted for publication in Nature Communications.

Response: We thank the reviewer for the appreciation of our revisions.